# Integrative genomic analysis of adult mixed phenotype acute leukemia delineates lineage associated molecular subtypes

Koichi Takahashi [1,2,3], Feng Wang[2], Kiyomi Morita[1], Yuanqing Yan[4], Peter Hu[5], Pei Zhao[5], Abdallah Abou Zhar[1], Chang Jiun Wu[2], Curtis Gumbs[2], Latasha Little[2], Samantha Tippen[2], Rebecca Thornton[2], Marcus Coyle[2], Marisela Mendoza[6], Erika Thompson[6], Jianhua Zhang[2,7], Courtney D. DiNardo[1], Nitin Jain[1], Farhad Ravandi[1], Jorge E. Cortes[1], Guillermo Garcia-Manero[1], Steven Kornblau[1], Michael Andreeff[1], Elias Jabbour[1], Carlos Bueso-Ramos[8], Akifumi Takaori-Kondo [3], Marina Konopleva[1], Keyur Patel[8], Hagop Kantarjian[1] & P. Andrew Futreal[2]

Mixed phenotype acute leukemia (MPAL) is a rare subtype of acute leukemia characterized by leukemic blasts presenting myeloid and lymphoid markers. Here we report data from integrated genomic analysis on 31 MPAL samples and compare molecular profiling with that from acute myeloid leukemia (AML), B cell acute lymphoblastic leukemia (B-ALL), and T cell acute lymphoblastic leukemia (T-ALL). Consistent with the mixed immunophenotype, both AML-type and ALL-type mutations are detected in MPAL. Myeloid-B and myeloid-T MPAL show distinct mutation and methylation signatures that are associated with differences in lineage-commitment gene expressions. Genome-wide methylation comparison among MPAL, AML, B-ALL, and T-ALL sub-classifies MPAL into AML-type and ALL-type MPAL, which is associated with better clinical response when lineage-matched therapy is given. These results elucidate the genetic and epigenetic heterogeneity of MPAL and its genetic distinction from AML, B-ALL, and T-ALL and further provide proof of concept for a molecularly guided precision therapy approach in MPAL.

[1] Department of Leukemia, The University of Texas MD Anderson Cancer Center, Houston, TX 77030, USA. [2] Department of Genomic Medicine, The University of Texas MD Anderson Cancer Center, Houston, TX 77030, USA. [3] Department of Hematology and Oncology, Graduate School of Medicine, Kyoto University, Kyoto 606-8397, Japan. [4] Department of Biostatistics, The University of Texas MD Anderson Cancer Center, Houston, TX 77030, USA. [5] Diagnostic Genetics Program, The University of Texas MD Anderson Cancer Center, Houston, TX 77030, USA. [6] Department of Genetics, The University of Texas MD Anderson Cancer Center, Houston, TX 77030, USA. [7] Institute of Applied Cancer Science, The University of Texas MD Anderson Cancer Center, Houston, TX 77030, USA. [8] Department of Hematopathology, The University of Texas MD Anderson Cancer Center, Houston, TX 77030, USA. These authors contributed equally: Koichi Takahashi, Feng Wang, P. Andrew Futreal. Correspondence and requests for materials should be addressed to K.T. (email: ktakahashi@mdanderson.org) or to P.A.F (email: afutreal@mdanderson.org)

Acute leukemia is a clonal hematopoietic malignancy that is characterized by increased proliferation and disorganized differentiation of hematopoietic cells. Although acute leukemia generally presents with either a myeloid or lymphoid lineage, rare cases present with blasts that show immunophenotypes of both myeloid and lymphoid lineages (biphenotypic) or with multiple blasts each having different lineage of immunophenotypes (bi-lineal). Such types of acute leukemia are classified as mixed phenotype acute leukemia (MPAL), which accounts for 1–3% of acute leukemias in adults[1]. The 2016 World Health Organization (WHO) classification of hematopoietic and lymphoid tumors defines five subtypes of MPAL: MPAL with t(9;22)(q34;q11.2), MPAL with MLL rearrangement, MPAL B/myeloid not otherwise specified (NOS), MPAL T/myeloid NOS, and MPAL NOS rare types[2]. Morphologically, the blasts of MPAL are indistinguishable from those of acute myeloid leukemia (AML) or acute lymphoid leukemia (ALL), and diagnosis relies on immunophenotyping. Because of the rarity of the disease and the mixed phenotype presentation, standard therapy for MPAL has not been clearly defined, leading to inconsistency in treatment choices between AML-directed regimens and ALL-directed regimens[3,4]. Patients with MPAL often present with abnormal karyotypes or complex chromosomal abnormalities, and their long-term prognosis is poor[3,5]. The underlying molecular pathophysiology that accounts for the mixed phenotype and distinct characteristics of MPAL is not well understood. Here, we performed multimodal molecular analyses of 31 adult MPAL samples using targeted-capture DNA sequencing, genome-wide methylation array, and RNA sequencing, with the purpose of delineating the genetic basis of MPAL and laying the groundwork for a precision therapy approach in MPAL. The analysis reveals the genetic and epigenetic heterogeneity of MPAL and potential link between molecular subtype and clinical response to therapy.

## Results

**Clinical characteristics of patients with adult MPAL.** The search of our institution's patient database identified 69 patients with the diagnosis of MPAL seen between 2000 and 2015. A total of 55 patients met the WHO diagnostic criteria for MPAL. Among those, 31 patients were untreated at the time of presentation and had pretreatment bone marrow samples available for analysis. The clinical characteristics of these 31 patients are summarized in Table 1. There was no significant difference in the characteristics between these 31 patients and 24 patients who were not eligible for the study because of prior therapy or non-availability of the samples (Supplementary Table 3). The median age of the studied cohort was 53 years (interquartile range: 30–61). Thirteen (42%) of the patients had an immunophenotype consistent with myeloid-B MPAL, while 18 (58%) had an immunophenotype-consistent myeloid-T MPAL. Karyotypes were abnormal in 21 of the 31 cases (68%), and 8 (26%) had complex karyotype abnormalities. Four cases of myeloid-B phenotype were positive for Philadelphia chromosome (Ph+) and one case of myeloid-T phenotype had 11q23 rearrangement (t[11;19][q23;p13.3]). The myeloid marker, myeloperoxidase (MPO), was detected in 85% of myeloid-B cases and 89% of myeloid-T cases. The cases in which MPO were not detected were positive for other myeloid markers such as non-specific esterase, CD11c, CD14, CD64, or lysozyme. All myeloid-T cases were positive for a T cell-specific markers (cytoplasmic CD3 or surface CD3), while all myeloid-B cases were positive for a B cell-specific marker (CD19). Most of the MPAL cases (87%) were positive for stem cell marker CD34, which was consistent with a previous report [3].

**Landscape of high-confidence somatic mutations in adult MPAL.** The targeted capture deep sequencing of 295 leukemia-enriched cancer genes revealed 65 high-confidence somatic single-nucleotide variants (SNVs) and 35 small insertions and deletions (indels) in 38 genes in 29 of 31 adult MPAL samples (94%) (Fig. 1a and Supplementary Table 2). The two cases with no detectable point mutations were Ph+ cases. The median number of cancer gene mutations was 2 (range: 0–7) per sample. The most frequently mutated genes were NOTCH1 in nine samples (29%, all myeloid-T), RUNX1 in eight samples (26%, six myeloid-B and two myeloid-T), and DNMT3A and IDH2 in seven samples each (23%, one myeloid-B and six myeloid-T for both). The myeloid-B and myeloid-T phenotypes showed distinct patterns of mutations (Fig. 1b), in which RUNX1 mutations were significantly enriched in myeloid-B ($P = 0.039$, odds ratio test), whereas NOTCH1 mutations showed significant enrichment in myeloid-T ($P = 0.029$, odds ratio test). Assessment of clonality based on the estimated cancer cell fraction (CCF) of the mutations revealed that RUNX1, DNMT3A, IDH2, SRSF2, KRAS, and TET2 mutations were frequently clonal, whereas mutations in FLT3, ETV6, TP53, and other rare genes were often subclonal or minimally subclonal (Fig. 1c, d).

We then compare the frequency of cancer gene mutations in MPAL to those of other lineage-committed acute leukemias (AML, B-ALL, and T-ALL) that were sequenced using the same platform (Fig. 1e). The number of detected mutations was similar between MPAL and AML (the median number of mutations was 2 [range: 0–7] in MPAL vs. 3 [range: 0–7] in AML, $P = 0.79$, Mann–Whitney $U$ test) or MPAL and T-ALL

| Table 1 Clinical characteristics of 31 patients with MPAL | | |
|---|---|---|
| | **Myeloid-B** | **Myeloid-T** |
| | **$N = 13$ (% or IQR)** | **$N = 18$ (% or IQR)** |
| Median age, years (IQR) | 59 (48–62) | 43.5 (30–60) |
| Female | 8 (61) | 5 (27) |
| Median WBC, ×10³/μL (IQR) | 4.05 (3.3–8.6) | 15.35 (2.2–18.4) |
| Median HGB, g/dL (IQR) | 8.95 (8.1–9.8) | 10.3 (8.6–11.8) |
| Median PLT, ×10³/μL (IQR) | 97 (27–143) | 52.5 (31–107) |
| Median BM blast percentage (IQR) | 66 (57–85) | 82 (62–89) |
| Median PB blast percentage (IQR) | 19 (11–43) | 60.5 (26–86) |
| Median LDH, IU/L (IQR) | 979 (598–1533) | 1012 (802–1208) |
| Cytogenetic abnormalities | | |
| Normal | 3 (23) | 7 (39) |
| Ph+ | 4 (31) | 0 (0) |
| 11q23 rearrangement | 0 (0) | 1 (6) |
| Complex | 4 (31) | 4 (22) |
| Other | 2 (15) | 6 (33) |
| Immunophenotype profile | N/tested | N/tested |
| MPO+[a] | 11/13 | 16/18 |
| CD3+ | 0/11 | 18/18 |
| CD10+ | 4/11 | 4/16 |
| CD11c+ | 2/2 | 3/3 |
| CD14+ | 0/15 | 1/18 |
| CD19+ | 13/13 | 0/18 |
| CD22+ | 11/12 | 1/13 |
| CD34+ | 13/13 | 14/18 |
| CD79a+ | 7/11 | 3/10 |

IQR interquartile range, WBC white blood cells, HGB hemoglobin, PLT platelets, BM bone marrow, PB peripheral blood, LDH lactate dehydrogenase, Ph+ Philadelphia chromosome positive, MPO myeloperoxidase
[a]Four cases with negative MPO were positive for more than two myeloid markers (CD11c+, CD14, non-specific estrase and lysozyme)

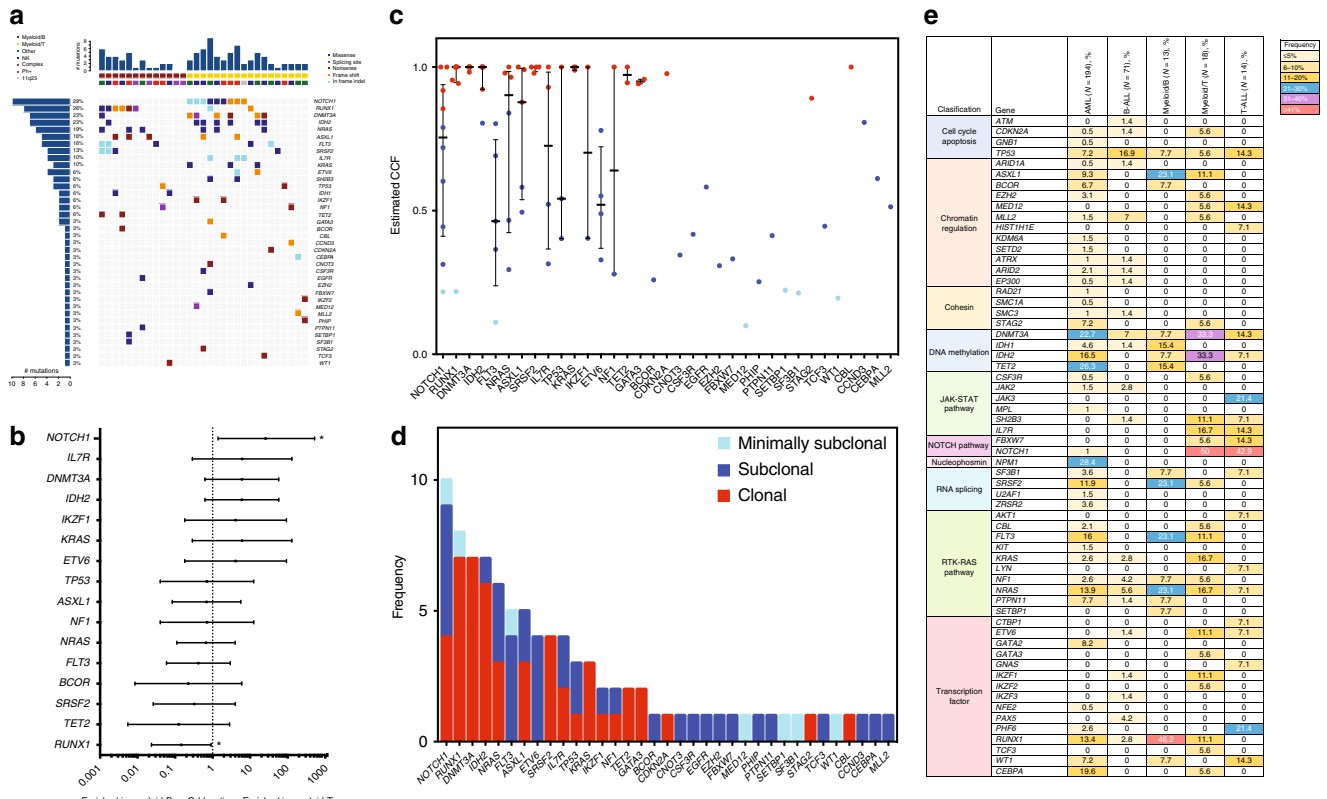

**Fig. 1** Landscape of somatic mutations in MPAL. **a** Landscape of high-confidence somatic mutations detected in 31 MPAL cases by sequencing with a 295-gene panel. Each column represents a case and each raw represents a gene. The top bar graph shows the number of mutations detected in each sample. The two rows directly underneath that graph show the (upper) immunophenotypes (myeloid-B in dark red and myeloid-T in yellow) and (lower) the cytogenetic classification. The bar graph at left shows the number of mutations detected overall in that gene. **b** Forrest plot showing enrichment of the mutations against the immunophenotypes by logarithmic odds ratio. *P < 0.05. The error bars represent 95% confidence interval of odds ratio. **c** Dot plot showing the estimated cancer cell fraction (CCF) of each mutation. Mutations with a CCF <0.2 are considered minimally subclonal (light blue), those with CCF ≥0.2 and <0.85 are considered subclonal (blue) and those with a CCF ≥0.85 are considered clonal (red). The error bars represent interquartile range and the center line represents the median. **d** Bar graph showing the number of detected mutations in each gene with degrees of clonality based on the estimated CCF. **e** Table summarizing the distribution of high-confidence somatic mutations detected by the 295-gene panel sequencing in patients with AML (N = 194), B-ALL (N = 71), myeloid-B MPAL (N = 13), myeloid-T MPAL (N = 18), or T-ALL (N = 14). The patients were classified by their diagnosis, and the mutations are grouped by the consensus molecular pathways. The frequency of the mutations is represented by different colors (key at upper right)

(the median number of mutations 2 [range: 0–7] in MPAL vs. 3 [range: 1–4] in T-ALL, P = 0.92, Mann–Whitney U test), but MPAL had significantly more mutations than B-ALL (the median number of mutations 2 [range: 0–7] in MPAL vs..0 [range: 0–4] in B-ALL, P < 0.001, Mann–Whitney U test). Mutations in *TP53*, *NRAS*, and *DNMT3A* were shared across all types of acute leukemia, albeit with some difference in frequency. In contrast, *NPM1* mutations were specific to AML and were not found in MPAL, which was consistent with a previous study[6]. Mutations in *IL7R*, *FBXW7*, and *NOTCH1* were specific to myeloid-T MPAL and T-ALL, suggesting a strong functional role of these genes in T cell lineage leukemia. While myeloid-T MPAL and T-ALL shared a number of mutations in common, there were also differences. *PHF6* and *JAK3* mutations, each detected in 21.4% of T-ALL cases, were not detected in myeloid-T MPAL (Fig. 1e). In contrast, *ASXL1* (11.1%) and *FLT3* (11.1%) mutations were detected in myeloid-T MPAL but not in T-ALL. Almost all of the mutations detected in myeloid-B MPAL were shared with AML, although the high prevalence of *RUNX1* mutations (46.2%) in myeloid-B MPAL is noteworthy (Fig. 1e). Overall, consistent with the mixed immunophenotype presentation, both AML-type and ALL-type mutations were detected in MPAL (Supplementary

Fig. 2). These data suggest that the aberrant immunophenotypes of MPAL may be partially driven by the mixture of both AML- and ALL-type mutations.

**DNA sequencing of longitudinal samples from MPAL patients.** Since all of the *DNMT3A* mutations detected in the current MPAL samples were non-R882 mutations (Supplementary Table 2) and were clonal in bone marrow (median estimated CCF = 1.0), we suspected that some MPAL cases are outgrowths of pre-leukemic clonal hematopoiesis, as clonal hematopoiesis is often associated with this type of *DNMT3A* mutations[7,8]. To address this question, we analyzed longitudinal bone marrow samples taken at the time of complete remission (CR) in eight MPAL cases, of which three had pretreatment *DNMT3A* mutations. As we expected, persistent *DNMT3A* mutations were detected in the CR samples in all three cases while other co-occurring mutations were cleared (Supplementary Fig. 3). These data suggest that the *DNMT3A* mutations detected in the MPAL cases are likely of pre-leukemic origin. We sequenced relapse samples available for two of the eight cases with longitudinal samples. In one of these cases (MDA016, myeloid-B), mutations in *GATA3* and *ARID2* were acquired at relapse. Interestingly, this

case lost MPO positivity at relapse, although the association between lineage loss and mutation acquisition is uncertain.

**Copy number alterations (CNAs) in MPAL.** Using data from the methylation array, we inferred genome-wide CNAs in the 31 MPAL samples (Supplementary Fig. 4). Recurrently detected arm-level CNAs were loss in chromosomes 5q (19%), 7 (6%), and 12p (13%), and gain in chromosome 4 (10%). Analysis of focal CNAs revealed copy number loss or deletions in genes that were often affected by loss-of-function mutations: *CDKN2A* (6%), *IKZF1* (3%), *FBXW7* (10%), *ETV6* (6%), and *CEBPA* (3%).

**Fusion transcripts in MPAL.** RNA sequencing data were available for 24 of the 31 patients (77%). Analysis of these data revealed, in addition to the translocation detected by karyotyping (*BCR-ABL1* [t(9;22)] and *MLL* rearrangement), five additional in-frame, non-recurrent fusion genes in three patients (Fig. 2). Case MDA020 (myeloid-T) had a *KMT2A* (*MLL*)-*MLLT4* fusion that was not apparent by karyotyping (cytogenetics showed normal karyotype), suggesting a cryptic abnormality. An *NSD1-NUP98* fusion was detected in one case of myeloid-B (MDA022), which has been described as a cryptic fusion in ~16 and 2% of pediatric and adult AML cases, respectively[9]. In addition to these two previously described leukemic fusions, we detected three additional in-frame fusions, *FOXP1-DNAJC15* and *TNKS-LYZ* in a myeloid-T case (MPAL27) and *NOP14-PLEC* in a myeloid-B case (MDA022). The pathogenic significance of these fusion transcripts is not clear. However, *FOXP1* translocations have been described previously in both lymphoid and myeloid leukemias (B-ALL[10,11] and myeloproliferative disorder[12]) with various partner genes such as *ABL1*, *PAX5*, and *PDGFR*, suggesting a role of this gene in both myeloid and lymphoid lineage leukemias. In a pediatric MPAL series, translocation of *ZNF384* was detected in ~40% of myeloid-B phenotype cases (personal communication with Dr. Charles Mulligan). We did not detect *ZNF384* fusion transcripts through our RNA sequencing. We also screened our cohort for the most common *ZNF384* fusions (*ZNF384-EP300*, *ZNF384-CREBBP*, *ZNF384-TAF15*, and *ZNF384-TCF3*) by RT-PCR but the fusions were not detected (Supplementary Fig. 5).

**DNA methylation and gene expression profiles in MPAL.** To further explore the potential molecular distinctions between the myeloid-T and myeloid-B phenotypes, we analyzed the genome-wide DNA methylation profile in the 31 MPAL samples. Consensus *k*-means clustering of the top 10,000 variably methylated probes identified two distinct clusters in MPAL (Fig. 3). These two methylation clusters showed significant correlation with immunophenotype: 73% of cases expressing cluster 1 were myeloid-B, whereas 88% of cases expressing cluster 2 were myeloid-T ($P < 0.001$). Overall, the myeloid-T cases had more hypermethylated CpG probes than the myeloid-B cases (Fig. 4), which was observed consistently among various CpG locations (Supplementary Fig. 6). Because *IDH1* and *IDH2* mutations were detected frequently in the current MPAL cohort (29%) and they are known to cause the CpG island hypermethylated phenotype (CIMP) in AML[13], we analyzed methylation differences between myeloid-T and myeloid-B phenotypes in *IDH1/2* wild-type and mutant cases separately. The methylation difference between the two phenotypes was still significant in *IDH1/2* wild-type cases, suggesting that this methylation difference is not conditional on *IDH1/2* mutation status (Supplementary Fig. 7).

We next analyzed promoter CpG probes that were differentially methylated between myeloid-T and myeloid-B phenotypes. Pathway analysis of differentially methylated CpG probes (DMP) revealed that promoter CpGs in genes associated with T cell receptor (TCR) signaling (*CD3D*, *PRKCQ*, *LCK*, and *CD247*) were hypo-methylated in myeloid-T cases (Fig. 5a). Integrated analysis of promoter methylation and mRNA expression data also revealed that TCR signaling genes, such as *CD3D*, *PRKCQ*, *CCR9*, and *CD7*, were differentially methylated and expressed between myeloid-T and myeloid-B cases (Fig. 5b, c). These data suggest that aberrant expression of CD3 in myeloid-T MPAL is mediated by promoter hypomethylation. We also performed motif enrichment analysis of promoter DMPs. This analysis revealed that *IRF8* and *IRF4* binding motifs were significantly hypo-methylated in myeloid-B cases compare to myeloid-T cases (Fig. 5d). Both *IRF8* and *IRF4* are transcription factors that play critical roles in pre-B cell development, differentiation, and function[14]. Consistent with the findings, *PAX5*, *CXCR4*, *RAG1*, and *RAG2*, downstream targets of *IRF4*/*IRF8* and essential to B cell development, were significantly upregulated in myeloid-B cases compare to myeloid-T cases (Fig. 5e). This is likely a mechanism of CD19 expression in myeloid-B MPAL, because

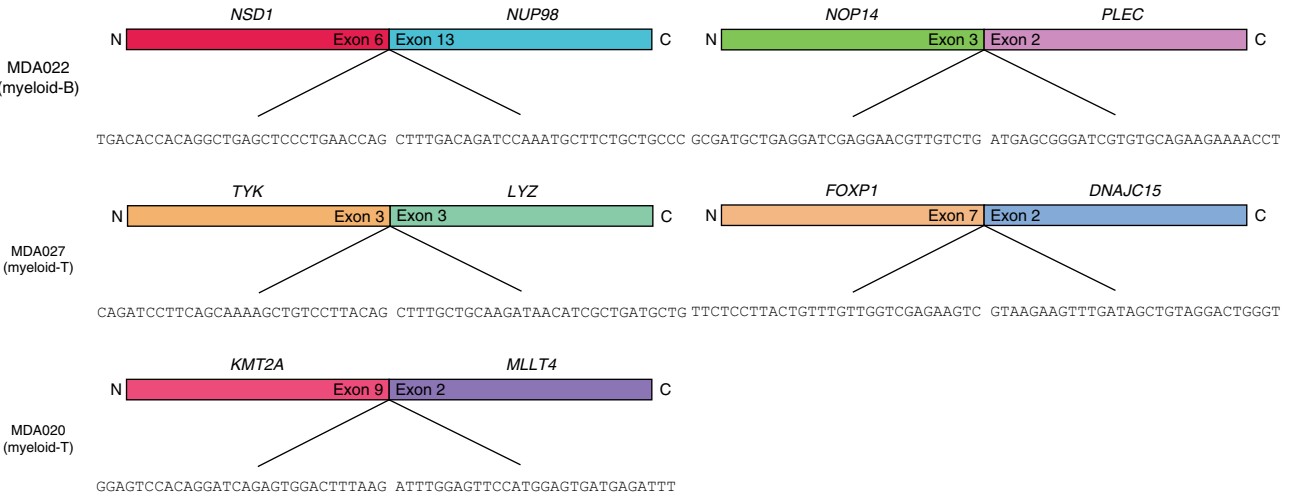

**Fig. 2** Fusion transcripts in MPAL. Fusion genes detected in MPAL samples by RNA sequencing. In addition to fusions already known (detected by karyotyping), a total five in-frame fusion candidates were detected in three patient samples

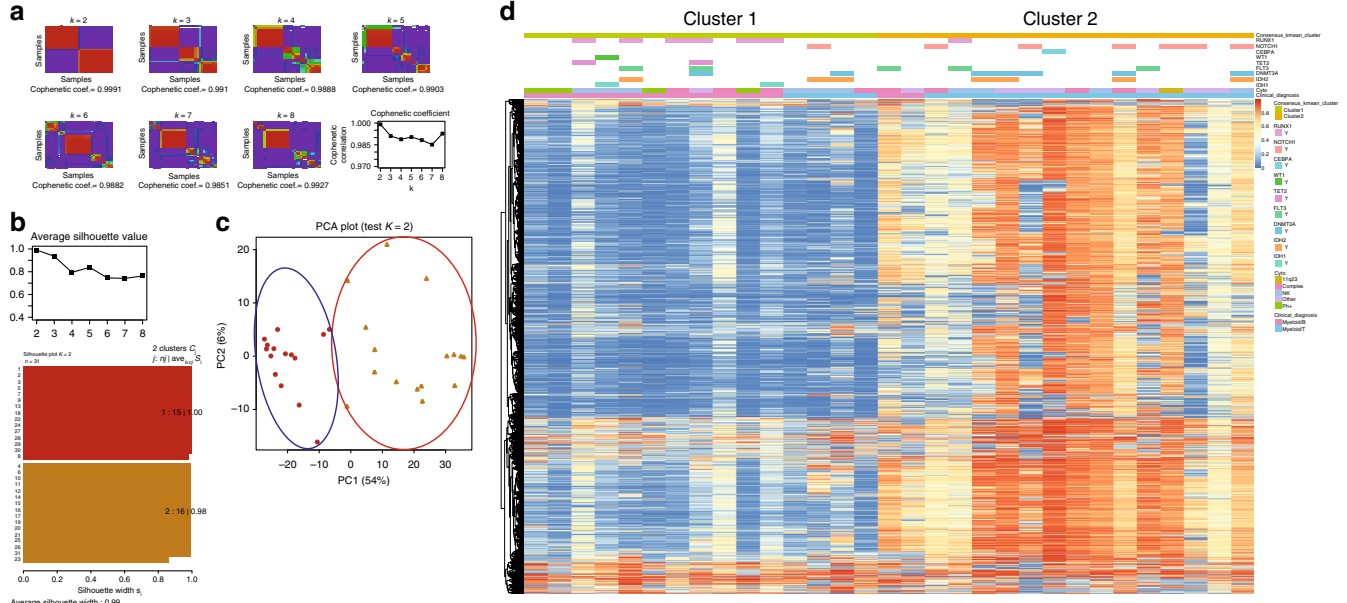

**Fig. 3** Methylation heterogeneity in MPAL. **a** Unsupervised consensus *k*-means clustering of the methylation data from 31 MPAL samples by the top 10,000 variably methylated probes. The consensus is the strongest at *k* = 2 with the highest cophenetic coefficient. **b** Silhouette analysis of individual *k* distrubution also supported that *k* = 2 has the highest average silhouette value. *j*: $n_j|$ ave$_{i \in cj}$ Si represents cluster number: number of samples grouped in the cluster| average silhouette width of the cluster. Silhouette distribution for other *k* values are also shown in Supplementary Fig. 9. **c** Principal component analysis (PCA) of MPAL samples based on 10,000 variably methylated probes supports 2 distinct clusters. **d** Unsupervised hierarchical clustering of the top 10,000 variably methylated probes with data on consensus clusters, genetic mutations, and immunosubtypes. Myeloid-B and myeloid-T phenotypes are significantly enriched in cluster 1 and cluster 2, respectively (*P* < 0.01)

PAX5 directly regulates the expression of CD19[15]. Additionally, gene set enrichment analysis (GSEA) of differentially expressed genes between the two phenotypes showed that the B cell receptor (BCR) and NF-κB pathways were upregulated in myeloid-B cases compare to myeloid-T cases (Fig. 5f). This is in line with the activation of the IRF4 pathway in the myeloid-B phenotype, because the BCR-NF-κB axis is an upstream inducer of *IRF4*[16]. Altogether, these data suggest that the immunophenotypic difference between myeloid-T and myeloid-B MPAL is affected substantially by differential activation of lineage-defining transcription factors and promoter methylation differences.

**Methylome comparison among MPAL and other acute leukemias**. We then compare the promoter CpG methylation patterns of AML, B-ALL, and T-ALL with that of MPAL to further explore differences between lineage-committed and mixed lineage leukemias. Unsupervised hierarchical clustering of the most variable promoter CpGs from AML, B-ALL, T-ALL, and MPAL cases differentiated AML, B-ALL, and T-ALL into distinct clusters (Fig. 6a). Because 18 of the 31 (59%) MPAL cases clustered with B-ALL or T-ALL and 13 (41%) clustered with AML, we designated these MPAL cases as ALL-like or AML-like MPAL, respectively (Fig. 6b). Of the 18 myeloid-T MPAL cases, 13 (72%) clustered with T-ALL, 5 (28%) clustered with AML, and none clustered with B-ALL. Of the 13 myeloid-B MPAL cases, 8 (62%) clustered with AML, 4 (31%) clustered with B-ALL, and 1 (7%) clustered with T-ALL. This case of myeloid-B MPAL that clustered with T-ALL had strong CD7 positivity along with CD19 and MPO positivity, which might explain why it clustered with T-ALL.

To better understand the difference between methylation-defined AML-like MPAL and ALL-like MPAL, we compare the mutation profiles and immunophenotypes of these two subtypes. Mutations in *RUNX1* and *SRSF2* were significantly associated with AML-like MPAL (Fig. 6c). Moreover, strong expression of

CD19 was associated with ALL-like myeloid-B MPAL (Fig. 6d) and strong association of CD7 with ALL-like myeloid-T MPAL (Fig. 6e).

**Correlation with clinical outcomes in MPAL**. Among the 31 MPAL patients analyzed in this study, 29 (94%) received induction chemotherapy (Supplementary Table 4). Of those, 17 (59%) received an ALL-directed regimen, 9 (31%) received an AML-directed regimen, and 3 (10%) received other types of therapy. Twelve of the 18 (67%) ALL-like MPAL cases were treated with an ALL-directed regimen, while 6 of 13 (46%) AML-like MPAL cases received AML-directed therapy. Among the 27 patients with evaluable response data, 18 received therapy that matched with their methylation-defined phenotype (i.e., AML-directed therapy for AML-like MPAL and ALL-directed therapy for ALL-like MPAL) and 9 patients received un-matched therapy. Notably, patients who received matched therapy were significantly more likely to achieve CR than patients who received un-matched therapy (CR rate: 72% vs. 22%, *P* = 0.037, Fisher's exact test, Fig. 7a). However, this did not translate to a difference in composite CR rate (CR + CR with insufficient blood recovery, Fig. 7a) and overall survival (Fig. 7b).

**Discussion**
In this study, we performed an integrative molecular analysis of 31 adult MPAL samples by using targeted gene sequencing, methylation array, and RNA sequencing. We also compare patterns of mutations and DNA methylation of MPAL to AML, B-ALL, and T-ALL to further define the molecular underpinnings of this unique clinical entity. Our data elucidated the genetic and epigenetic heterogeneity associated with the distinct myeloid-B and myeloid-T immunophenotypes of MPAL. While the myeloid-B and myeloid-T phenotypes shared some cancer gene mutations in common, the pattern of somatic mutations differed

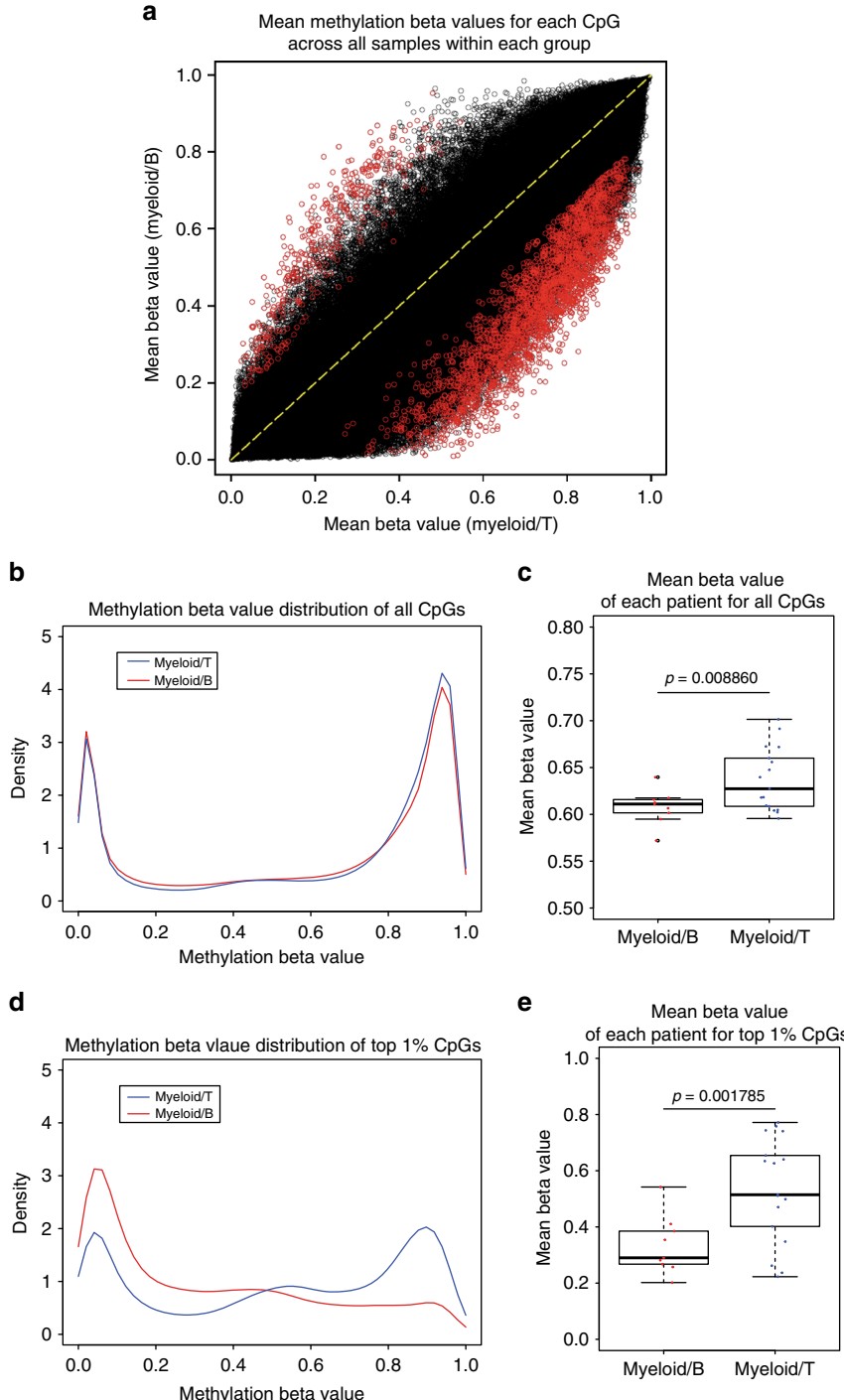

**Fig. 4** Methylation difference between myeloid-T and myeloid-B MPAL. **a** Scatter plot with mean beta value of each CpG probe among all 31 MPAL samples compared between myeloid-B (*y* axis) and myeloid-T (*x* axis) phenotypes. Red-colored dots represent probes that were significantly differentially methylated (DMP: FDR <0.1 and difference in mean delta value >0.15) between the two phenotypes. The myeloid-T phenotype has more DMPs with higher methylation beta values than the myeloid-B phenotype. **b** Density distribution of all CpG probes with methylation beta values comparing myeloid-B and myeloid-T phenotypes. **c** Box plot comparing mean methylation beta value of all CpG probes in each MPAL sample based on immunophenotypes. **d** Density distribution of top 1% variably methylated probes with methylation beta values comparing myeloid-B and myeloid-T phenotypes. **e** Box plot comparing mean methylation beta values of top 1% variably methylated probes in each MPAL sample based on immunophenotypes. For box plots, center line represents the meidan, box edges represent the 25th and 75th percentiles, upper whisker represent 75th percentile + 1.5 times interquartile range, lower whisker represent 25th percentile − 1.5 times interquartile range and the dots represent actual data point

between the two phenotypes, with significant differences in the frequency of *NOTCH1* and *RUNX1* mutations. Furthermore, myeloid-B and myeloid-T MPAL cases showed distinct DNA methylation patterns that were associated with differences in expression of key genes essential to hematopoietic lineage commitment. In comparison with AML, B-ALL, and T-ALL, the

mutation landscape and methylation pattern were similar between myeloid-T MPAL and T-ALL although there were some differences. For example, myeloid-T MPAL lacked *JAK3* and *PHF6* mutations which were detected in ~20% of T-ALL. Additionally, based on the RNA sequencing, we did not detect oncogenic transcription factor fusions that are often

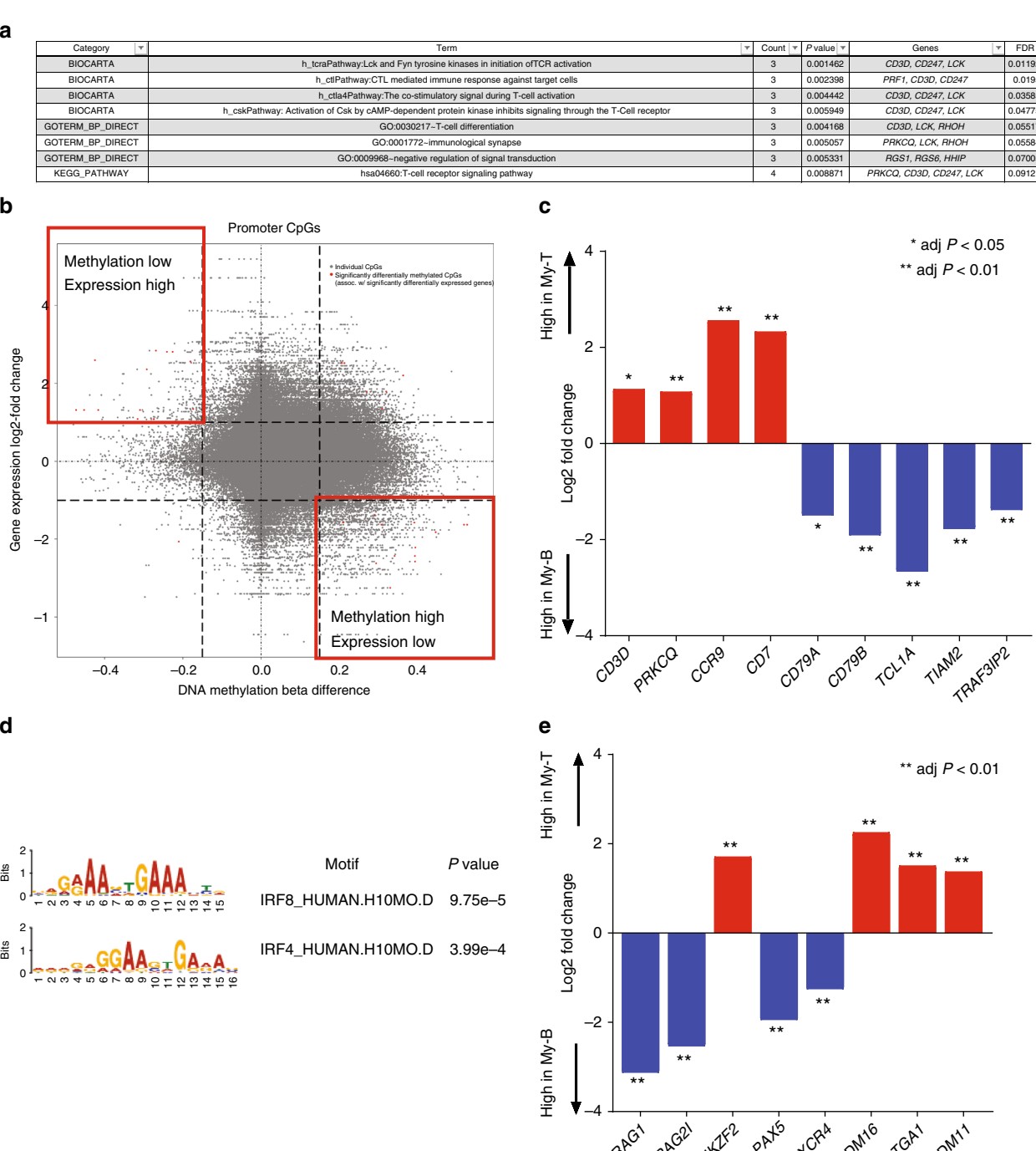

characteristic to T-ALL such as but not limited to *TAL1*, *TLX1*, *TLX3*, *NKX2.1*, *LMO1*, and *LMO2* fusions. These differences might represent the unique molecular feature of MPAL.

*DNMT3A* mutations were frequently detected in myeloid-T MPAL (33%) and all mutations were nonsense or frameshift mutations with high VAF leading to loss of function (Supplementary Table 2). This is similar to the mutation characteristics in adult T-ALL[17]. Furthermore, loss-of-function *DNMT3A* mutations have been shown to cooperate with NOTCH1 activation in T-ALL mice model[18]. In fact, in our cohort, four of six (67%) *DNMT3A* mutated myeloid-T had co-occurrence of *NOTCH1* mutation, supporting the cooperative mechanism between these two genes in T cell lineage malignancy. Interestingly, there was a strong co-occurrence of *DNMT3A* and *IDH2* mutations (five of six DNMT3A-mutated cases had IDH2 mutations) in myeloid-T, which is not seen in T-ALL and maybe unique to myeloid-T MPAL. As a result of this co-occurrence, we found no difference in global methylation level between *DNMT3A* mutated and wild-type myeloid-T cases (Supplementary Fig. 8), because *IDH2* and *DNMT3A* have been shown to have epigenetic antagonism[19]. One of the significant DMPs between *DNMT3A* mutated and wild-type myeloid-T cases was the CpGs in gene body of *NOTCH1* (Supplementary Fig. 8) and these loci overlapped with one of the permissive enhancers detected in FANTOM project[20]. Functional consequence of this hypo-methylated enhancer region is unclear and we did not find difference in gene expression of *NOTCH1* by *DNMT3A* mutation.

The clonal origin of MPAL and the mechanism underlying its mixed-phenotype presentation have been debated since the first-case description of MPAL[4,21,22]. One theory, "lineage promiscuity", refers to a mechanism whereby leukemic transformation occurs at the stage of early hematopoietic stem/progenitor cells (HSC/HSPC) with multi-lineage potential, allowing promiscuous expression of myeloid and lymphoid lineage markers[23]. Another theory, "lineage infidelity", argues that oncogenic mutations misconfigure differentiation program of leukemia cells, leading to the mixed phenotype presentation[24]. Although our study was not designed to provide a definitive answer to this controversy, our data suggest that these two theories may not be mutually exclusive and that both mechanisms may be in play for MPAL development. Frequent positivity of early stem cell markers, CD34 and CD117, and the fact that MPAL is most often myeloid-T or myeloid-B, not T-B, support the lineage promiscuity concept that MPAL arises from early HSC/HSPCs that have multi-lineage potential[3]. At the same time, our mutation data support the lineage infidelity mechanism, in that oncogenic mutations in lineage-defining transcription factor genes such as *NOTCH1* and *RUNX1* were detected frequently in MPAL. *RUNX1* mutations were previously reported to have significant association with minimally differentiated AML (M0 by the French-American-British [FAB] classification), and analysis of gene expression profiles showed that *RUNX1*-mutated AML

had significant upregulation of the BCR pathway[25–27]. This is in line with our data indicating that myeloid-B MPAL, 40% of which had *RUNX1* mutations, had BCR-NFκB pathway activation, which likely resulted in IRF4-PAX5 activation and subsequent CD19 expression. Similarly, activating mutations in *NOTCH1* and *IL7R* in myeloid-T MPAL may explain the aberrant expression of T cell markers[28]. Taken together, our results suggest a model of MPAL development in which HSCs/HSPCs with multi-lineage potential acquire mutations in lineage-defining transcription factor genes that facilitate expression of the mixed immunophenotype. The putative candidate cell of origin is the multi-lymphoid progenitor (MLP) because it has bi-potential capacity to develop into lymphoid (B and T) and myelomonocytic lineages[29]. It is still unknown whether myeloid-T and myeloid-B phenotypes arise from the same clonal origin. The significant differences in methylation signatures between the two phenotypes, which was independent of *IDH* mutations, raises the possibility that the two phenotypes inherited distinct methylation signatures from different cell types, because cell-specific methylation signatures are generally well preserved during hematopoietic differentiation and can be used as a clonal fingerprint[30]. A further analysis of the mutation mapping in purified HSC compartments from MPAL patients might help in identifying the precise clonal origin.

Clinically, MPAL often generates diagnostic and therapeutic challenges[4]. No standard therapy has been defined for MPAL, and treatment choice has been inconsistent between AML-type and ALL-type regimens. Previous retrospective studies of MPAL suggested that an ALL-type induction regimen produces higher response and survival rates[3,5]. This is consistent with our data, because the majority of the MPAL cases in our cohort were classified as ALL-like MPAL based on methylation profile, and thus an ALL-type regimen may render a higher response rate overall. However, an ALL-type regimen might not be recommended for all MPAL patients, because 40% of MPAL showed a methylation pattern similar to that of AML. In our retrospective analysis, patients who received therapy "matched" to their methylation profile had significantly higher rates of CR to induction therapy than patients who received "un-matched" therapy. These data serve as proof of concept for the molecularly guided precision therapy approach in MPAL. Currently, genome-wide methylation analysis is not widely implemented in the clinic, but a recent study showed the potential feasibility and reproducibility of methylation assays in the clinical setting[31]. Future studies in a larger cohort of MPAL patients correlating clinical profiles and outcomes with molecular analysis, especially methylation analysis, are needed to confirm our results. Another clinically relevant finding from this study is that ~50% of MPAL patients carried at least one clinically actionable mutation (*IDH2* and *FLT3*). Midostaurin (FLT3 inhibitor) and enasidenib (IDH2 inhibitor) are clinically effective in AML with *FLT3* and *IDH2* mutations, respectively[32,33]. The clinical activity of

---

**Fig. 5** Transcriptomic changes between myeloid-T and myeloid-B. **a** Pathway analysis of differentially methylated promoter CpG probes (FDR <0.1 and delta beta value >0.15) between myeloid-B and myeloid-T phenotypes showing significant enrichment of T cell receptor pathways. *P* value was calculated by Fisher's exact test and FDR was calculated by Benjamini–Hochsberg method. **b** Starburst plot integrating analysis of gene expression and promoter methylation. Red dots represent promoter CpG probes with significantly differential methylation that are also associated with significant differences in expression between myeloid-B and myeloid-T. **c** Log2 fold differences of transcription levels of genes that were significantly different between myeloid-B and myeloid-T phenotypes and were associated with significant promoter methylation differences. *P* value was calculated by Wald test and adjusted for multiple testing by Benjramini–Hochberg method. **d** Motif enrichment analysis of promoter CpG probes differentially methylated between the two phenotypes showing significant enrichment of *IRF8* and *IRF4* recognition motifs. *P* value was calculated by Wilcoxon rank-sam test. **e** Log2 fold differences of transcription levels of key downstream target genes of *IRF8* and *IRF4* between myeloid-B (My-B) and myeloid-T (My-T) phenotypes. *P* value was calculated by Wald test and adjusted for multiple testing by Benjramini–Hochberg method. **f** Gene set enrichment analysis (GSEA) comparing gene expression data from RNA sequencing between myeloid-B and myeloid-T phenotypes showing significant enrichment of B cell receptor (BCR) and NFκB pathways in myeloid-B MPAL. The method of estimating nominal *P* value and FDR adjustment is described elsewhere [52]

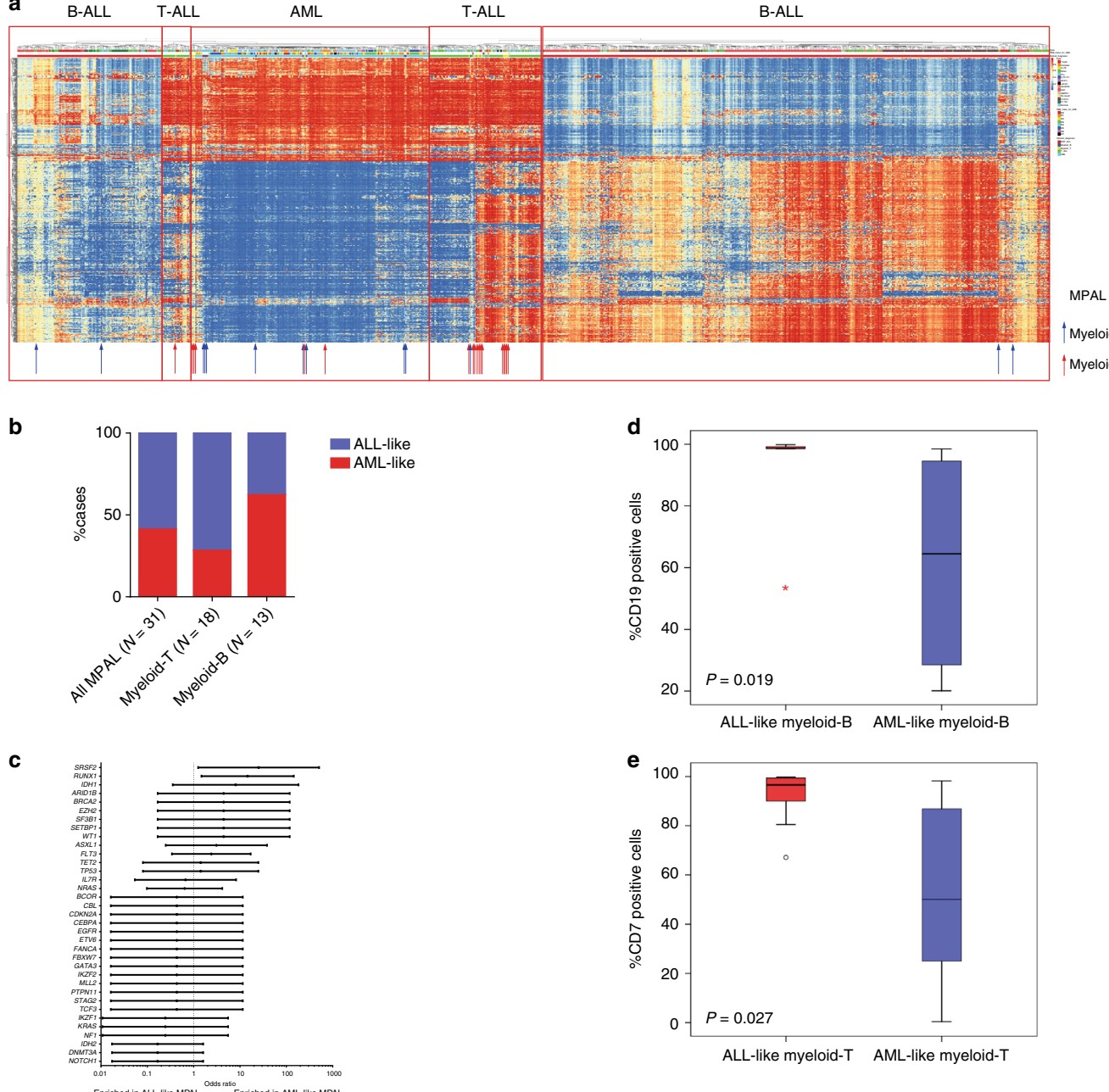

**Fig. 6** Methylation comparison among all leukemias. **a** Unsupervised hierarchical clustering of the top 10,000 variably methylated probes among AML (N = 194), B-ALL (N = 505), T-ALL (N = 101), and MPAL (N = 31) cases. MPAL cases are indicated by arrows (red arrow, myeloid-T: blue arrow, myeloid-B). MPAL cases that were clustered within AML cases and B-ALL/T-ALL cases were classified as "AML-like MPAL" and "ALL-like MPAL," respectively. **b** Distribution of AML-like MPAL and ALL-like MPAL defined by methylation cluster in myeloid-T and myeloid-B MPAL. **c** Forest plot showing enrichment of mutations against AML-like MPAL and ALL-like MPAL by logarithmic odds ratio. The error bars represent 95% confidence interval of odds ratio. **d**, **e** Box plots showing frequency of CD19-positive and CD7-positive cells in myeloid-B and myeloid-T MPAL, respectively, stratified by ALL-like and AML-like MPAL. For box plots, center line represents the meidan, box edges represent the 25th and 75th percentiles, upper whisker represent 75th percentile + 1.5 times interquartile range, lower whisker represent 25th percentile − 1.5 times interquartile range and the dots represent outliers. Difference between the two groups was tested by Mann–Whitney U test

these agents in MPAL is not known, but its important MPAL cases are screened for these clinically actionable mutations for potential inclusion of these patients in molecularly targeted agent trials.

This study has several limitations. First, the sample size is small and therefore some of the analysis has limited statistical power to draw a definitive conclusion. For example, *DNMT3A*, *IDH2*, and *FLT3* were more frequently mutated in myeloid-T compare to

myeloid-B but the difference was not statistically significant. We might identify more immunophenotype-specific mutations if the cohort is large. Given the rarity of the disease, this is a limitation of a retrospective study from a single center and it highlights the need of a multi-institutional large study. Second, due to the lack of normal control, we did not perform whole exome/genome DNA sequencing and instead performed targeted DNA sequencing. Although our targeted sequencing panel covered 295 genes

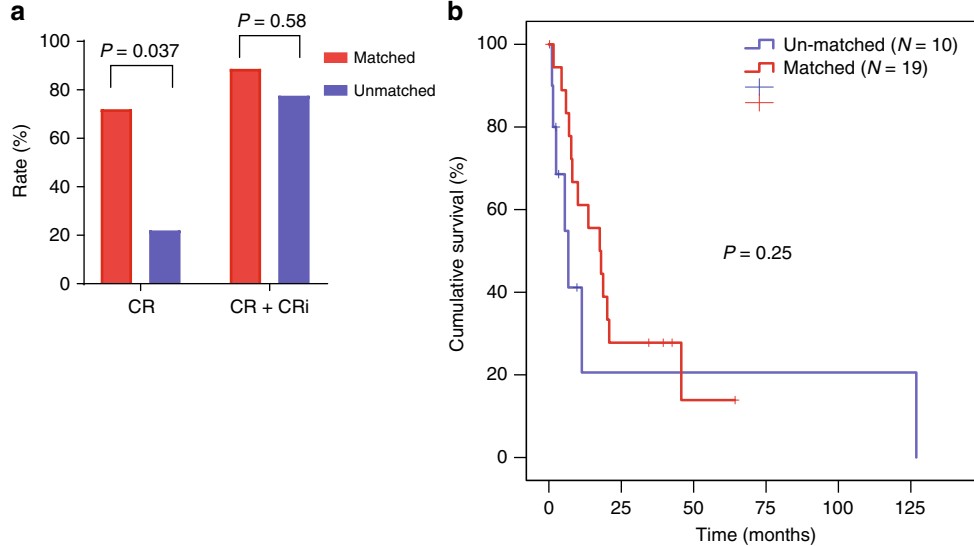

**Fig. 7** Clinical correlation. Comparison of **a** complete remission (CR) and **b** composite CR (CR + CR with insufficient count recovery [CRi]) rate in MPAL patients after 1 cycle of induction chemotherapy based on whether the therapy was matched with the patient's MPAL methylation status. Difference between two groups was tested by Fisher's exact test. **c** Comparison of overall survival in patients who received a therapy matched to their methylation status and in patients who received an un-matched therapy. Difference between two grous was tested by log-rank test

that are recurrently mutated in both myeloid and lymphoid malignancies, there is a chance that missed novel MPAL genes. Third, an interpretation of the results on methylation-defined subtype and its correlation with clinical response/outcome needs caution because of the small sample size and variability in post-induction therapy.

In summary, we report data from an integrated molecular analysis of 31 adult MPAL cases. To the best of our knowledge, this is the most comprehensive and detailed molecular analysis of MPAL. Our data delineate distinct genetic and epigenetic bases for myeloid-T and myeloid-B MPAL phenotypes and suggest potential molecular mechanisms for the mixed immunophenotype presentation.

## Methods

**Patients**. We searched our institution's medical record database for patients with acute leukemia evaluated between 2000 and 2015; from that subset, we identified patients whose disease was diagnosed as MPAL. Because MPAL cases diagnosed before 2008 were classified as "bi-phenotypic leukemia" according to the European Group of Immunological Markers for Leukemia (EGIL) criteria[34], we reclassified those diagnoses according to the 2008 WHO criteria[35]. Also, for the purpose of this study, we included only MPAL cases with a bi-phenotypic presentation; bi-lineal cases were excluded because of the presumed biological and clonal differences of these two diseases. This resulted in 55 patients meeting the diagnostic criteria of MPAL by 2008 WHO Classification. We then selected patients whose MPAL was untreated at the time of presentation at MD Anderson and who had pretreatment bone marrow samples available for analysis. For some of these MPAL patients, bone marrow samples taken at the time of complete remission (CR) and/or relapse were available for longitudinal analysis. Mutation frequencies in these samples were compare with those in samples from patients with other types of acute leukemia, including previously untreated AML (N = 194), precursor B cell ALL (pre-B-ALL, N = 71), and T cell ALL (T-ALL, N = 14). Written informed consent for sample collection and analysis was obtained from all patients. The study protocols adhered to the Declaration of Helsinki and were approved by the Institutional Review Board at The University of Texas MD Anderson Cancer Center.

**Targeted gene next-generation sequencing and mutation calling**. We used a SureSelect custom panel of 295 genes (Agilent Technologies, Santa Clara, CA) that are recurrently mutated in hematologic malignancies (Supplementary Table 1). This panel was designed to cover a wide range of recurrently mutated genes in both myeloid and lymphoid malignancies, making it suitable for the analysis of mixed phenotype leukemia. Details of the sequencing methods have been described previously[36]. Briefly, genomic DNA was extracted from diagnostic bone marrow

aspirates, as well as CR and relapse specimens when available, using an Autopure extractor (QIAGEN/Gentra, Valencia, CA). DNAs were fragmented and bait-captured in solution according to the manufacturer's protocols. Captured DNA libraries were then sequenced using a HiSeq 2000 sequencer (Illumina, San Diego, CA) with 76 base pair paired-end reads.

**Bioinformatics analysis calling high-confidence somatic mutations**. The bioinformatic pipelines calling high-confidence somatic single-nucleotide variants (SNVs) and indels from targeted capture DNA sequencing were described previously[37]. Cancer cell fraction (CCF) of each mutation was estimated by a series of formulas. Briefly, raw ploidy was calculated from the copy number segmentation data, adjusted by tumor purity, and rounded to the next integer. The rounded-adjusted ploidy, along with raw variant allele frequency and tumor purity, were then used to estimate the CCF. Detailed formulas are listed below. Diploid status was assumed. Mutations in sex chromosomes were skipped and only adjusted by tumor purity.

$$Raw\_ploidy = 2 \times 2^{**}cnv\_log2\_ratio$$

$$Adjusted\_ploidy = (Raw\_ploidy - 2 + (2 \times Purity))/Purity$$

Adjusted_ploidy is rounded to the next integer.

$$Adjusted\_VAF = Raw\_VAF \times (Purity \times Rounded\_adjusted\_ploidy + 2 \times (1 - Purity))/Purity/2$$
$$CCF = Adjusted\_VAF \times 2$$

**DNA methylation analysis and data processing**. DNA methylation analysis was performed using Illumina's Infinium MethylationEPIC assay (EPIC), which covers ~850,000 CpG positions in the human genome. Briefly, genomic DNA from bone marrow samples was treated with sodium bisulfite (Zymo Research, Irvine, CA) that coverts un-methylated cytosine residues to uracil and then processed by whole-genome amplification, enzymatic fragmentation, and hybridization to bead chips at 48 °C for 17 h according to the manufacturer's protocol. After array hybridization, specific probes designed to interrogate bisulfite-converted loci were single-base extended by incorporating DNP or biotin-labeled ddNTP and stained with a fluorescent reagent to determine the signal intensity ratio of methylated versus un-methylated residues. After washing and staining, the bead chips were scanned on iScan (Illumina) to generate IDAT files. We used the ChAMP algorithm for data processing and normalization following the program's default procedures[38]. The IDAT files were taken as input files and raw beta values were generated. Following initial filtering and quality check, the data were normalized using the BMIQ method[39]. The batch effect was assessed by using the singular value decomposition (SVD) method and corrected if necessary[40]. Differential methylation analysis was performed by using the limma algorithm[41]. For comparison of methylation status among AML, B-ALL, T-ALL, and MPAL,

methylation data from 194 AML samples were downloaded from The Cancer Genome Atlas Data Portal (https://tcga-data.nci.nih.gov/docs/publications/laml_2012/) and methylation data from 505 B-ALL and 101 T-ALL samples analyzed for a previously published study were kindly provided by Drs. Syvanen and Nordlund from Uppsala University[42,43]. Raw IDAT files from both data sources were taken as raw input files and processed and normalized using the same pipeline already described. Since both of these data sets were generated by Illumina's previous platform, the Infinium HumanMethylation450 (HM450) array that covered ~485,000 CpG positions, we analyzed only the CpG probes that overlapped between EPIC and HM450 (91% of the HM450 probes overlapped with the EPIC probes, Supplementary Fig. 1).

**RNA sequencing and data processing.** cDNA was synthesized via Ribo-SPIA Technology (NuGEN, San Carlos, CA) per the manufacturer's instructions. The synthesized cDNA library was sequenced by Illumina's HiSeq 2000 platform. Raw sequencing data from the Illumina platform were converted to fastq files and aligned to the reference genome (hg19) using the Spliced Transcripts Alignment to a Reference (STAR) algorithm[44]. HTSeq-count was then utilized to generate the raw counts for each gene[45]. Raw counts were then analyzed by DESeq2 for data processing, normalization, and differential expression analysis according to standard procedures[46]. To identify candidate fusion transcripts, we ran three different structural variation (SV) detection algorithms: TopHat-Fusion[47], FusionMap[48], and MapSplice[49]. SVs that met the following criteria were considered candidate fusion transcripts: (1) SVs supported consistently by 2 or more algorithms, (2) in-frame fusion transcripts, (3) reads that were not mapped to different transcripts by Blat search, (4) supporting seed reads count ≥4, and (5) junction located at the exon-intron boundaries.

**Estimation of copy number alterations (CNAs).** Genome-wide CNAs were estimated from methylation array data using a combination of the Conumee algorithm (http://bioconductor.org/packages/conumee/) and our in-house segmentation pipeline. Briefly, IDAT files were loaded into Conumee and combined intensity values were generated and normalized. The raw log2 ratio for each CpG site was then calculated by comparing against a set of eight normal internal controls. The calculated log2 ratios were subjected to segmentation by the circular binary segmentation method[50]. Ploidy value was calculated from the segmentation log2 ratios and then adjusted by tumor purity that was estimated from bone marrow blast percentage. The final adjusted ploidy was then plotted by using the Copynumber R package [51].

**Gene set enrichment analysis.** Normalized counts data from DESeq2 was taken as input for Gene Set Enrichment Analysis (GSEA)[52]. The permutation parameter was set as the gene set, while the remaining parameters were kept as the defaults.

**Motif enrichment analysis.** We defined differentially methylated CpG probes (DMP) with a difference between myeloid-T and myeloid-B phenotypes corresponding to a false discovery rate (FDR) value <0.1 and a delta beta value >0.15. The list of DMPs and non-DMPs (as controls) was then mapped to the genome and the nearest transcription start site (TSS) was identified. Genomic sequences 2000 bp upstream and downstream of the TSS were extracted using BEDTools[53]. Extracted sequences were then taken as input for the motif enrichment analysis using the Analysis of Motif Enrichment (AME) software [54].

**Reverse-transcriptase polymerase chain reaction (RT-PCR).** Total RNA was isolated from bone marrow aspiration samples and were reverse transcribed to generate complementary DNA (cDNA) using iScript cDNA Synthesis Kit, (Bio-Rad). We screened for various ZNF384 fusions using the following primer pairs. EP300-ZNF384: Forward primer TCTAGGGGTGGGTCAACAGT, Reverse primer CTGTCAGCAAGGTGGGGTAG, TCF3-ZNF384: Forward primer CAGCCTCATGCACAACCAC Reverse primer CCAGTGTGGATTCGTGTGTG, CREBBP-ZNF384: Forward primer CTCTCGGACTCCCCTACATGA Reverse primer TCAGCAAGGTGGGGTAGTGA, TAF15-ZNF284: Forward primer GGAAGCCAAGGTGGAAGAG Reverse primer ACAGCCCTTCTCTGGCAAC. To complement the lack of positive control, we used two internal controls. First to validate the RT-PCR procedure by confirming expression of GAPDH: Forward primer GAGTCAACGGATTTGGTCGT Reverse primer TTGATTTTGGAGGGATCTCG. Second to confirm expression of ZNF384: Forward primer AACCCTTCAAGTGCCACAAC Reverse primer GCACCTGTTGCTGAAGATCA.

**Statistical analysis.** The chi-square or Fisher's exact test was used to assess differences in categorical variables, and the Mann–Whitney $U$ test or Student's $t$ test was used to analyze differences in continuous variables. Survival outcomes were plotted by the Kaplan–Meier method and the difference in survival was assessed by the log-rank test. Statistical analyses were performed using SPSS (version 24; IBM Corporation, Armonk NY) and R (ver. 3.1.3).

**Code availability.** We used publicly available computer codes to generate results and all codes are cited in the method.

**Data availability.** De-identified mutation data and clinical data are available in the Supplementary Information. The data sets generated from RNA sequencing and methylation array are available in the Gene Expression Omnibus repository with the following accession numbers: GSE113601 (RNA sequencing) and GSE113545 (methylation array).

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

## Acknowledgements

This study was supported in part by the Cancer Prevention Research Institute of Texas (grant R120501 to P.A.F.), the Welch Foundation (grant G-0040 to P.A.F.), the UT System STARS Award (grant PS100149 to P.A.F.), the Red and Charline McCombs Institute for the Early Detection and Treatment of Cancer Award (to K.T.), an Institutional Research Grant at the MD Anderson Cancer Center (to K.T.), Khalifa Scholar Award (to K.T.), Charif Souki Cancer Research Fund (to H.K.), MD Anderson Cancer Center Leukemia SPORE P50 CA100632 (to H.K.), MD Anderson Cancer Center Support Grant (NIH P30 CA016672), and by generous philanthropic contributions to MD Anderson's Moon Shot Program (P.A.F.). We thank Kathryn Hale at Department Scientific Publications at MD Anderson for providing scientific editing of the manuscript.

## Author contributions

K.T. designed the study, analyzed data, and wrote the manuscript. F.W., C.J.W., and J.Z. performed the bioinformatics analysis. K.M. analyzed the data and wrote the manuscript. Y.Y. performed the statistical analysis. P.Z., P.H., C.B.R., and K.P. performed the pathological analysis. A.A.Z. collected the data. C.G., L.L., S.T., R.T., and M.C. performed DNA sequencing and RNA sequencing. M.M. and E.T. performed the methylation analysis. A.T.-K. critically reviewed the manuscript. C.D.D., N.J., F.R., J.C., G.G.M., S.K., M.A., H.K., E.J., M.K., and K.P. collected samples and treated the patients. P.A.F. provided leadership, managed the study team, and wrote the manuscript. All authors read and approved the manuscript.

## Additional information

**Competing interests:** The authors declare no competing interests.

