## [Peer Review File · Nature Communications]

Reviewers' comments:

Reviewer #1 (Remarks to the Author):

This is a very detailed genomic/transcriptomic analysis of adult mixed phenotype acute leukaemia (MPAL).

Major comments:

1. One limitation of the study is that only targeted sequencing has been used to define the spectrum of mutations in MPAL. Although it is likely that most mutations in MPAL are mutations that are known in ALL or AML, it is also possible that new genes could be mutated specifically in MPAL cases. Therefore, it would be of interest to also perform exome sequencing. Since there are only 31 cases in the study, that is not too difficult to do with current technology. This is not essential, but would make the story complete.

2. T-ALL is characterised by the presence of chromosomal rearrangements leading to ectopic expression of TAL1, TLX1, TLX3, NKX2.1, LMO1 or MLO2. Were any of these typical T-ALL translocations observed in the MPAL cases with T-immunophenotype? This could be identified on RNA-seq data, and would possibly be another factor discriminating true T-ALL from myeloid T-MPAL if such translocations are absent in myeloid T-MPAL.

Reviewer #2 (Remarks to the Author):

This paper describes an examination of MPAL, a relatively rare form of mixed-phenotype leukemia having poor prognosis. This is an important and timely topic for advanced molecular analysis, especially since MPAL cannot be distinguished morphologically from other leukemias. The investigators discuss their multipronged approach that integrates DNA and RNA sequencing and methylation profiling, as well as their bioinformatics processing and analyses of these data. They also describe biomedically important results, including distinct mutation patterns and methylation signatures in the phenotypes and sub-classifications having differences in clinical responses. This work advances scientific progress regarding MPAL and would therefore be of interest to the readership of the journal.

The paper is well written, but there are scientific aspects that must be clarified or expanded in order to give full details to the reader. Several are minor. For example, singular value decomposition for handling batch effect (page 5) should have a reference (there are several with minor variations). Details for how purity was estimated in correcting for ploidy (page 7) are missing. Also on page 7, the statement "adjusted P value < 0.1" appears. Do the authors mean FDR, for which 0.1 would be reasonable, or a single P value, for which 0.1 is too high? There are also several issues on which readers would appreciate additional commentary by the investigators, mainly related to limitations of the study. First, to what degree might the 295 gene panel be missing details that might appear in larger WES or even WGS examinations? I suspect that added information would not change the primary observations appreciably, especially since mutations were only found in ~40 genes, but it would be helpful to have the investigators' views. Also, the cohort size of 31 is limited, although this is quite understandable, given the rarity of this form of the disease. It would be interesting to hear the investigators' thoughts on what effects a larger cohort having more power might have. For example, is it likely that other gene mutations would become more definitively associated with Myeloid-B and Myeloid-T, as RUNX1 and NOTCH1 are respectively for the current data set? Lastly, the observation that statistically significant differences in complete remission rates between matched therapy versus unmatched therapy cases did not result in

a difference in overall survival seems a little surprising. Is this a reflection of limited statistical power?

Finally, there are a few methodological issues that should be further explained. Regarding the DNA methylation analysis using K-means clustering in Figure 3(A), the topology of the clustering seems to depend significantly on the choice of K. For example, there are significant differences for K = 2 through K = 6. The investigators seem to prefer the K = 2 result, e.g. Figure 3(B), based on its having the highest cophenetic coefficient, 0.999. However, the lowest coefficient within the set is 0.985 for K = 7, which is not significantly different. It would be important to provide additional analyses to address this issue.

Reviewer #3 (Remarks to the Author):

This is a nice study of the genomics of mixed phenotype acute leukemia (MPAL). Though a relatively rare disease, it is a challenging entity from a clinical standpoint and very interesting from a scientific perspective. The authors report an extensive and well-performed genomic analysis of a relatively small number of samples (though more than have been reported previously). This reveals a number of interesting conclusions.

Specific points:

1. Further analysis and discussion of DNMT3A mutations would be interesting – the mutation distribution (not R882), lineage specificity, and VAF are all interesting points. Were there DNMT3A-specific methylation changes?
2. A main point of the paper was matching treatment to subtype. But there is actually quite a limited amount done with the data and limited clinical information. No real info was provided on treatments, number of cycles, reason for remission failure, or outcome with subsequent treatment. Nor is there any info on other differences between the treatment groups that could confound the findings.
3. How many were lost because of prior treatment or lack of material and were these demographically different?

Reviewers' comments:

Reviewer #1 (Remarks to the Author):

This is a very detailed genomic/transcriptomic analysis of adult mixed phenotype acute leukaemia (MPAL).

Major comments:

1. One limitation of the study is that only targeted sequencing has been used to define the spectrum of mutations in MPAL. Although it is likely that most mutations in MPAL are mutations that are known in ALL or AML, it is also possible that new genes could be mutated specifically in MPAL cases. Therefore, it would be of interest to also perform exome sequencing. Since there are only 31 cases in the study, that is not too difficult to do with current technology. This is not essential, but would make the story complete.

RE: We completely agree with the reviewer that whole exome/genome sequencing is an ideal DNA sequencing method to thoroughly evaluate a genomic abnormality of MPAL. The main reason why we chose targeted DNA sequencing on this project was because of the lack of matched normal control. Without having matched normal control we thought that WES or WGS approach would yield significantly high false-positive results and difficulty in distinguishing somatic mutations from germline polymorphisms. Unfortunately, these MPAL cases are rare and therefore we had to rely on specimens that were stored in the past. And most of the patients we analyzed are unfortunately deceased making it very difficult to obtain matched germline control at this point. Previously, Eckstein et al. performed WES in 23 MPAL samples (Eckstein, Wang et al. 2016). Matched normal control was also not available in their study and thus the mutation landscape reported in their paper was not particularly different from ours. This illustrates the limitation of discovery potential by WES/WGS if matched normal control is not available. Our targeted sequencing panel was designed to cover a wide range of genes that have been reported to be recurrently mutated in both myeloid and lymphoid malignancies. Therefore we hope that this panel covered major driver genes in MPAL. We fully agree, though, that this is a major limitation of our study and there may be un-identified MPAL genes. We acknowledged the limitation in our revised manuscript.

Page 13, Line 16-25.

“This study has several limitations. First, the sample size is small and therefore some of the analysis has limited statistical power to draw a definitive conclusion. For example, DNMT3A, IDH2, and FLT3 were more frequently mutated in myeloid-T compared to myeloid-B but the difference was not statistically significant. We might identify more immunophenotype-specific mutations if the cohort is large. Given the rarity of the disease, this is a limitation of a retrospective study from a single center and it highlights the need of a multi-institutional large study. Second, due to the lack of normal control, we did not perform whole exome/genome DNA sequencing and instead performed targeted DNA sequencing. Although our targeted sequencing panel covered 295 genes that are recurrently mutated in both myeloid and lymphoid malignancies, there is a chance that missed novel MPAL genes. Third, an interpretation of the results on methylation-defined subtype and its correlation with clinical response/outcome needs caution because of the small sample size, cross activity of ALL and AML directed regimen, and variability in post-induction therapy.”

2. T-ALL is characterised by the presence of chromosomal rearrangements leading to ectopic

expression of *TAL1*, *TLX1*, *TLX3*, *NKX2.1*, *LMO1* or *LMO2*. Were any of these typical T-ALL translocations observed in the MPAL cases with T-immunophenotype? This could be identified on RNA-seq data, and would be possibly be another factor discriminating true T-ALL from myeloid T-MPAL if such translocations are absent in myeloid T-MPAL.

RE: Thank you for this comment. The reviewer rightly points out that T-ALL has characteristic translocations involving genes such as *TAL1*, *TLX1*, *TLX3*, *NKX2.1*, *LMO1* or *LMO2*. Interestingly, we did not find any of these T-ALL specific translocations in our MPAL, more specifically in myeloid-T cases. To rule out the possibility that our fusion callers have filtered these fusions, we analyzed raw unfiltered fusion calls and specifically looked for translocations involving these genes. We found 1 fusion involving *TAL1* but it only had 1 split read supporting the fusion and partner sequence had multiple homology sequence and therefore this fusion call was considered low confidence. We also found another call involving *LMO2* and it involved exon-intron boundaries but only had 2 seed counts and did not pass our confidence criteria. We agree with the reviewer that lack of these characteristic translocations might be one of the discriminating factors between T-ALL and myeloid-T MPAL. We added this discussion in the revised manuscript.

Page 10 and line 17-23.

“In comparison with AML, B-ALL and T-ALL, the mutation landscape and methylation pattern were similar between myeloid-T MPAL and T-ALL although there were some differences. For example, myeloid-T MPAL lacked *JAK3* and *PHF6* mutations which were detected in approximately 20% of T-ALL. Additionally, based on the RNA sequencing, we did not detect oncogenic transcription factor fusions that are often characteristic to T-ALL such as but not limited to *TAL1*, *TLX1*, *TLX3*, *NKX2.1*, *LMO1* and *LMO2* fusions. These differences might represent the unique molecular feature of MPAL.”

Reviewer #2 (Remarks to the Author):

This paper describes an examination of MPAL, a relatively rare form of mixed-phenotype leukemia having poor prognosis. This is an important and timely topic for advanced molecular analysis, especially since MPAL cannot be distinguished morphologically from other leukemias. The investigators discuss their multipronged approach that integrates DNA and RNA sequencing and methylation profiling, as well as their bioinformatics processing and analyses of these data. They also describe biomedically important results, including distinct mutation patterns and methylation signatures in the phenotypes and sub-classifications having differences in clinical responses. This work advances scientific progress regarding MPAL and would therefore be of interest to the readership of the journal.

The paper is well written, but there are scientific aspects that must be clarified or expanded in order to give full details to the reader. Several are minor. For example, singular value decomposition for handling batch effect (page 5) should have a reference (there are several with minor variations).

RE: We apologize that we missed to cite a reference here. The new reference was added.

New reference #12 “Teschendorff et al. Genome Research 2010”

Details for how purity was estimated in correcting for ploidy (page 7) are missing.

RE: Thank you. We estimated purity from bone marrow blast percentage and added this clarification.

Page 17 and line 19.

“Ploidy value was calculated from the segmentation log₂ ratios and then adjusted by tumor purity **that was estimated from bone marrow blast percentage.**”

Also on page 7, the statement "adjusted P value < 0.1" appears. Do the authors mean FDR, for which 0.1 would be reasonable, or a single P value, for which 0.1 is too high?

RE: Indeed the P value was adjusted by Benjamini-Hochsberg method and therefore it represents FDR. The change is reflected in the revised manuscript.

Page 18 and line 2-4.

“We defined differentially methylated CpG probes (DMP) with a difference between myeloid-T and myeloid-B phenotypes corresponding to a false discovery rate (FDR) value < 0.1 and a delta beta value > 0.15.”

There are also several issues on which readers would appreciate additional commentary by the investigators, mainly related to limitations of the study. First, to what degree might the 295 gene panel be missing details that might appear in larger WES or even WGS examinations? I suspect that added information would not change the primary observations appreciably, especially since mutations were only found in ~40 genes, but it would be helpful to have the investigators' views.

RE: We absolutely agree with this. May we refer the reviewer to our comments in response to the reviewer #1's first comment.

Also, the cohort size of 31 is limited, although this is quite understandable, given the rarity of this form of the disease. It would be interesting to hear the investigators' thoughts on what effects a larger cohort having more power might have. For example, is it likely that other gene mutations would become more definitively associated with Myeloid-B and Myeloid-T, as RUNX1 and NOTCH1 are respectively for the current data set?

RE: We thank the reviewer for this comment. We totally agree with the reviewer that the small sample size of this study limit the statistical power of the analysis. We acknowledged this limitation in the revised manuscript.

Page 13 and line 16-22.

“This study has several limitations. First, the sample size is small and therefore some of the analysis has limited statistical power to draw a definitive conclusion. For example, DNMT3A, IDH2, and FLT3 were more frequently mutated in myeloid-T compared to myeloid-B but the difference was not statistically significant. We might identify more immunophenotype-specific mutations if the cohort is large. Given the rarity of the disease, this is a limitation of a retrospective study from a single center and it highlights the need of a multi-institutional large study.”

Lastly, the observation that statistically significant differences in complete remission rates between matched therapy versus unmatched therapy cases did not result in a difference in overall survival seems a little surprising. Is this a reflection of limited statistical power?

RE: Thank you for this remark. As the reviewer rightly pointed out, the small sample size of the study is likely one of the reasons that response to therapy and survival did not correlate. Another reason is the heterogeneity in post-induction therapy. In response to the reviewer #3's comment, we added detailed information on clinical courses and it demonstrates high variability in post induction therapy, stem cell transplant status, and salvage therapies. These likely confounded the long-term survival data. In addition, although methylation-defined subtype correlated with CR rate, it did not have statistical correlation with composite CR rate (CR+ Cri, P = 0.58, new **Figure 7A**). With these caveat, we emphasized in the revised manuscript that the interpretation of this correlation analysis needs caution.

Page 10 line 4.

However, this did not translate to a difference in **composite CR rate (CR + CR with insufficient blood recovery, Figure 7A)** and overall survival (Figure 7B).

Page 14, line 1-2.

“Third, an interpretation of the results on methylation-defined subtype and its correlation with clinical response/outcome needs caution because of the small sample size and variability in post-induction therapy.”

Finally, there are a few methodological issues that should be further explained. Regarding the DNA methylation analysis using K-means clustering in Figure 3(A), the topology of the clustering seems to depend significantly on the choice of K. For example, there are significant differences for K = 2 through K = 6. The investigators seem to prefer the K = 2 result, e.g. Figure 3(B), based on its having the highest cophenetic coefficient, 0.999. However, the lowest coefficient within the set is 0.985 for K = 7, which is not significantly different. It would be important to provide additional analyses to address this issue.

RE: Thank you for this feedback. To help identifying the best clustering of our methylation data, we also calculated silhouette average value along with cophenetic coefficient and both analyses supported the use of K=2 in this analysis. We added the results of Silhouette analysis to the revised manuscript (**Figure 3B and Supplemental Figure S9**).

k	Cophenetic coefficient	Average silhouette width
2	0.9991	0.990405
3	0.991	0.935856
4	0.9888	0.794007
5	0.9903	0.838545
6	0.9882	0.745952

7	0.9851	0.741543
8	0.9927	0.762909

Reviewer #3 (Remarks to the Author):

This is a nice study of the genomics of mixed phenotype acute leukemia (MPAL). Though a relatively rare disease, it is a challenging entity from a clinical standpoint and very interesting from a scientific perspective. The authors report an extensive and well-performed genomic analysis of a relatively small number of samples (though more than have been reported previously). This reveals a number of interesting conclusions.

Specific points:

1. Further analysis and discussion of *DNMT3A* mutations would be interesting – the mutation distribution (not R882), lineage specificity, and VAF are all interesting points. Were there *DNMT3A*-specific methylation changes?

RE: Thank you for this suggestion. Although statistically not significant, *DNMT3A* mutations were more frequently detected in myeloid-T than in myeloid-B. Plus, they were all non-R882 mutations, which is in line with the *DNMT3A* mutation distribution for T-ALL (Grossmann, Haferlach et al. 2013). These non-R882 mutations have been shown to have loss of function effect (while R822 has been shown to have dominant negative effect) and they have been shown to cooperate with *NOTCH1* to accelerate T-ALL in mice model (Kramer, Kothari et al. 2017). The median VAF of *DNMT3A* mutations were 0.72 (IQR: 0.46-0.84) suggesting that some leukemia cells carry homozygous or heterozygous *DNMT3A* mutations with LOH. This also mirrors the findings in T-ALL.

To better understand the functional role of *DNMT3A* mutations in myeloid-T MPAL, we analyzed DMR in *DNMT3A* mutated myeloid-T MPAL compared to WT cases. Interestingly, there was no significant overall methylation difference by *DNMT3A* mutation. This is likely due to the fact that almost all *DNMT3A* mutated cases had co-occurring *IDH2* mutations in the current myeloid-T MPAL. In AML, a previous study showed that co-occurrence of *DNMT3A* and *IDH2* have been shown to have antagonistic effect, leading to a neutral methylation state (Glass, Hassane et al. 2017). Although there was no global methylation difference by *DNMT3A* mutation, we analyzed DMR in *DNMT3A* mutated cases. One of the statistically significant DMR (beta value difference more than 0.15 and FDR < 0.1) involved CpGs in the body of *NOTCH1*, which were significantly hypo-methylated in *DNMT3A* mutated cases compared to *DNMT3A* wild type cases. Part of the DMR overlapped with one of the passive enhancers identified in FANTOM project (Genehancer ID: GH09H136524). The enhancer has not been functionally investigated and functional consequence of this hypo-methylation at the enhancer region in *DNMT3A* mutated cases is not clear. At the very least, we did not find difference in RNA expression level of *NOTCH1* itself between *DNMT3A* mutated and WT cases. We added these data as **Supplemental Figure S8** and added discussion in the revise manuscript.

Page 10-11.

“*DNMT3A* mutations were frequently detected in myeloid-T MPAL (33%) and all mutations were nonsense or frameshift mutations with high VAF leading to loss-of-function (Supplemental Table 2). This is similar to the mutation characteristics in adult T-ALL.(Grossmann, Haferlach et al. 2013) Furthermore, loss-of-function *DNMT3A* mutations have been shown to cooperate with

NOTCH1 activation in T-ALL mice model.(Kramer, Kothari et al. 2017) In fact, in our cohort, 4 of 6 (67%) *DNMT3A* mutated myeloid-T had co-occurrence of *NOTCH1* mutation, supporting the cooperative mechanism between these 2 genes in T-cell lineage malignancy. Interestingly, there was a strong co-occurrence of *DNMT3A* and *IDH2* mutations (5 of 6 *DNMT3A* mutated cases had *IDH2* mutations) in myeloid-T, which is not seen in T-ALL and maybe unique to myeloid-T MPAL. As a result of this co-occurrence, we found no difference in global methylation level between *DNMT3A* mutated and wild type myeloid-T cases (Supplemental Figure S8), because *IDH2* and *DNMT3A* have been shown to have epigenetic antagonism.(Glass, Hassane et al. 2017) One of the significant DMPs between *DNMT3A* mutated and wild type myeloid-T cases was the CpGs in gene body of *NOTCH1* (Supplemental Figure S8) and these loci overlapped with one of the permissive enhancers detected in FANTOM project.(Andersson, Gebhard et al. 2014) Functional consequence of this hypomethylated enhancer region is unclear and we did find difference in gene expression of *NOTCH1* by *DNMT3A* mutation.”

2. A main point of the paper was matching treatment to subtype. But there is actually quite a limited amount done with the data and limited clinical information. No real info was provided on treatments, number of cycles, reason for remission failure, or outcome with subsequent treatment. Nor is there any info on other differences between the treatment groups that could confound the findings.

RE: Thank you for this feedback. We added more detailed clinical information including the details on induction treatments, consolidation therapy, response to therapy, bone marrow/stem cell transplantation and salvage therapies. This information was added as **Supplemental table S4**.

3. How many were lost because of prior treatment or lack of material and were these demographically different?

RE: Among the 55 MPAL cases we identified from the database search, 24 cases were removed from the study. Among those, 9 cases had previously untreated MPAL, but either baseline samples were not available or the patients did not consent for sample storage protocol. Other 15 patients received prior therapy before the presentation to our institution and therefore were removed from the study. We compared clinical characteristics of 31 MPAL patients studied to those of 24 patients removed from the study (**Supplemental Table S3**). There was no significant difference in the clinical characteristics between the two groups.

References

1. Andersson, R., C. Gebhard, I. Miguel-Escalada, I. Hoof, J. Bornholdt, M. Boyd, Y. Chen, X. Zhao, C. Schmidl, T. Suzuki, E. Ntini, E. Arner, E. Valen, K. Li, L. Schwarzfischer, D. Glatz, J. Raithel, B. Lilje, N. Rapin, F. O. Bagger, M. Jorgensen, P. R. Andersen, N. Bertin, O. Rackham, A. M. Burroughs, J. K. Baillie, Y. Ishizu, Y. Shimizu, E. Furuhashi, S. Maeda, Y. Negishi, C. J. Mungall, T. F. Meehan, T. Lassmann, M. Itoh, H. Kawaji, N. Kondo, J. Kawai, A. Lennartsson, C. O. Daub, P. Heutink, D. A. Hume, T. H. Jensen, H. Suzuki, Y. Hayashizaki, F. Muller, A. R. R. Forrest, P. Carninci, M. Rehli and A. Sandelin

- (2014). "An atlas of active enhancers across human cell types and tissues." Nature **507**(7493): 455-461.
2. Eckstein, O. S., L. Wang, J. N. Punia, S. M. Kornblau, M. Andreeff, D. A. Wheeler, M. A. Goodell and R. E. Rau (2016). "Mixed-phenotype acute leukemia (MPAL) exhibits frequent mutations in DNMT3A and activated signaling genes." Exp Hematol **44**(8): 740-744.
 3. Glass, J. L., D. Hassane, B. J. Wouters, H. Kunimoto, R. Avellino, F. E. Garrett-Bakelman, O. A. Guryanova, R. Bowman, S. Redlich, A. M. Intlekofer, C. Meydan, T. Qin, M. Fall, A. Alonso, M. L. Guzman, P. J. M. Valk, C. B. Thompson, R. Levine, O. Elemento, R. Delwel, A. Melnick and M. E. Figueroa (2017). "Epigenetic Identity in AML Depends on Disruption of Nonpromoter Regulatory Elements and Is Affected by Antagonistic Effects of Mutations in Epigenetic Modifiers." Cancer Discov **7**(8): 868-883.
 4. Grossmann, V., C. Haferlach, S. Weissmann, A. Roller, S. Schindela, F. Poetzinger, K. Stadler, F. Bellos, W. Kern, T. Haferlach, S. Schnittger and A. Kohlmann (2013). "The molecular profile of adult T-cell acute lymphoblastic leukemia: mutations in RUNX1 and DNMT3A are associated with poor prognosis in T-ALL." Genes Chromosomes Cancer **52**(4): 410-422.
 5. Kramer, A. C., A. Kothari, W. C. Wilson, H. Celik, J. Nikitas, C. Mallaney, E. L. Ostrander, E. Eultgen, A. Martens, M. C. Valentine, A. L. Young, T. E. Druley, M. E. Figueroa, B. Zhang and G. A. Challen (2017). "Dnmt3a regulates T-cell development and suppresses T-ALL transformation." Leukemia **31**(11): 2479-2490.

REVIEWERS' COMMENTS:

Reviewer #1 (Remarks to the Author):

The authors have responded to all questions/remarks. I have no further comments.

Reviewer #2 (Remarks to the Author):

I have no additional comments.

Reviewer #3 (Remarks to the Author):

The authors have answered all reviewer comments well.

REVIEWERS' COMMENTS:

Reviewer #1 (Remarks to the Author):

The authors have responded to all questions/remarks. I have no further comments.

Reviewer #2 (Remarks to the Author):

I have no additional comments.

Reviewer #3 (Remarks to the Author):

The authors have answered all reviewer comments well.

RE: We thank all reviewers' for their effort and careful review of our paper.